# Metabolic Syndrome Screening and Nutritional Status of Patients with Psoriasis: A Scoping Review

**DOI:** 10.3390/nu15122707

**Published:** 2023-06-10

**Authors:** Nur Hanisah Mohamed Haris, Shanthi Krishnasamy, Kok-Yong Chin, Vanitha Mariappan, Mohan Arumugam

**Affiliations:** 1Dietetics Program, Faculty of Health Sciences, Universiti Kebangsaan Malaysia, Jalan Raja Muda Abdul Aziz, Kuala Lumpur 50300, Malaysia; 2Centre for Diagnostic, Therapeutic and Investigative Studies, Faculty of Health Sciences, Universiti Kebangsaan Malaysia, Jalan Raja Muda Abdul Aziz, Kuala Lumpur 50300, Malaysia; 3Department of Pharmacology, Faculty of Medicine, Universiti Kebangsaan Malaysia, Bandar Tun Razak, Cheras, Kuala Lumpur 56000, Malaysia; 4Centre for Toxicology and Health Risk Studies, Faculty of Health Sciences, Universiti Kebangsaan Malaysia, Jalan Raja Muda Abdul Aziz, Kuala Lumpur 50300, Malaysia; vanitha.ma@gmail.com; 5Internal Medicine & Dermatology, Faculty of Medicine, Universiti Kebangsaan Malaysia, Bandar Tun Razak, Cheras, Kuala Lumpur 56000, Malaysia; a.mohan@ukm.edu.my

**Keywords:** plaque psoriasis, scoping review, metabolic syndrome, nutritional assessment, nutritional status

## Abstract

*Background:* Patients with plaque psoriasis have an increased risk of metabolic syndrome. However, no studies have assessed the nutritional status or screening methods of this population. *Aims:* This review aimed to identify and summarise metabolic syndrome screening criteria and the tools/methods used in nutrition assessment in patients with plaque psoriasis. *Data synthesis*: PubMed, Web of Science, Ovid and Scopus were searched from inception to March 2023, following the Arkensey and O’Malley framework, to identify articles that report nutritional assessment methods/tools and metabolic screening criteria. Twenty-one studies were identified. Overall, these studies used four different screening criteria to define metabolic syndrome. Patients with psoriasis had a high prevalence of metabolic syndrome and had a poor nutritional status compared to controls. However, only anthropometric measures such as weight, height and waist circumference were employed to determine the nutritional status. Only two studies assessed the vitamin D status. *Conclusions:* Patients with psoriasis have a poor nutritional status, and they are at risk of nutrient deficiencies. However, these health aspects are not routinely assessed and may increase the risk of malnutrition among these patients. Therefore, additional assessments, such as body composition and dietary assessment, are needed to determine the nutritional status to provide a suitable intervention.

## 1. Introduction

Plaque psoriasis or psoriasis vulgaris is a chronic inflammatory autoimmune skin disease affecting 60 million people worldwide [1,2]. It is characterised by erythematous scaly patches covering large amounts of the skin, mainly on the extensor surfaces, such as the elbows and knees, as well as the scalp, trunk and gluteal fold [3]. Disease severity using the Psoriasis Area and Severity Index (PASI) score is widely used in research, but not routinely, to assess skin involvement, location, thickness, scaling and redness [4]. The pathogenesis of psoriasis involves genetic predisposition, a dysregulated immune response, inflammatory pathways and environmental factors such as obesity and nutrition [5,6,7]. Patients with psoriasis have an increased risk of developing cardiovascular diseases, type 2 diabetes, metabolic syndrome (MetS), obesity, autoimmune thyroid disease, gout, mental health diseases, gastrointestinal diseases, chronic kidney diseases and even malignancy [8], leading to an economic burden, poor quality of life and reduced productivity [9,10].

MetS is the most common comorbidity in patients with psoriasis, with a prevalence rate of 20–50% [11]. In a clinical setting, physicians often use one of these screening criteria distributors to diagnose MetS: the World Health Organization (WHO) [12], the National Cholesterol Education Program-Adult Treatment Panel III (NCEP-ATP III) [13] and the European Group for the Study of Insulin Resistance (EGIR) [14,15]. The main clinical feature of MetS is insulin resistance but measuring circulating insulin is not routine in clinical practice. The common surrogate measures for insulin resistance and its complications are waist circumference for abdominal adiposity, triglycerides (TG) and high-density lipoprotein (HDL) cholesterol levels for dyslipidaemia, fasting blood glucose for diabetes mellitus and blood pressure for hypertension [16]. The mechanism linking psoriasis and MetS is still unclear. Th1 and Th17 T-cell-mediated inflammation are associated with the expression of pro-inflammatory cytokines, such as IL-6 and TNF-alpha, which enter the systemic circulation, inducing metabolic disorders such as insulin resistance and endothelial dysfunction, thus increasing the risk of diabetes and cardiovascular diseases [17,18]. Meanwhile, abdominal adiposity in patients with psoriasis induces inflammation by mediating pro-inflammatory cytokines and adipokines such as leptin and resistin, contributing to the development of insulin resistance, dyslipidaemia and vascular dysfunction [19] (Figure 1).

All these measured parameters form part of the nutritional assessment, which is routinely performed in clinical practice by dietitians to identify nutritional problems or risks to optimise nutritional management and determine the nutritional status of an individual. Other measures include anthropometrical indices, biochemical evaluation, clinical evaluation and food and diet history. Measurement of anthropometry includes the weight, body mass index (BMI), skinfold measurement and body composition. However, recently, body composition using a bioimpedance analysis (BIA), dual X-ray absorptiometry (DXA) and computed tomography (CT) has been used to measure adiposity [20]. Meanwhile, biochemical parameters that are usually assessed include the complete blood count, electrolytes and liver parameters [21,22]. Clinical evaluation determines a patient’s inability to chew or swallow, loss of appetite, metabolic stress due to infection, gastrointestinal symptoms such as vomiting, diarrhoea and nausea, fluid retention and clinical signs on the skin indicating micronutrient deficiency, as well as functional assessment to determine muscle function [22]. Finally, the assessment of diet history involves food and beverage intake and the influence of cultural or religious factors, also taking into account allergies, food preference, supplement intake and intolerance to estimate total energy, carbohydrate, fat and protein intake [21]. Therefore, a comprehensive nutrition assessment is pertinent to determine the nutritional status of patients.

Although patients with psoriasis with MetS are more likely to be over-nourished due to obesity, the excess energy intake may be mainly from calorie-dense low-nutrient foods. A Brazilian study among male patients with psoriasis and patients with psoriasis arthritis aged between 19 and 60 years reported a high intake of calories, total fat and protein exceeding the recommended levels and a lower intake of fibre and minerals [23]. These results were similar to an Italian study assessing seven days of dietary intake whereby patients had a higher carbohydrate, total fat and polyunsaturated fatty acid (PUFA) intake and lower fibre intake compared to controls [24]. Meanwhile, in a Turkish case-control study, an association between low vitamin E intake and severity in patients was observed [25]. Although patients had higher BMI compared to controls, their overall intake was significantly lower compared to controls. It is evident that patients with psoriasis, particularly those with MetS, have a poor nutritional status and poor-quality diet that may lead to nutritional deficiencies. It is important for patients attending a clinic to complete a thorough nutritional assessment. Therefore, this scoping review was undertaken to identify and summarise current findings on the types of nutrition assessment in patients with psoriasis in clinical settings. This review also identifies knowledge gaps in planning treatment and relevant research in the future.

## 2. Methods

This scoping review was conducted following the framework outlined by Arkensey and O’Malley [26]. Five different steps were followed based on the framework: (1) identifying the research question; (2) identifying relevant studies; (3) selecting studies; (4) charting the data; and (5) collating, summarising and reporting results. Reporting of this review was in accordance with the Preferred Reporting Items for Systematic Reviews and Meta-Analyses Extension for Scoping Reviews (PRISMA-ScR) [27].

### 2.1. Identifying the Research Question

The research question was the following: what tools/methods are used for nutrition assessment in patients with psoriasis?

### 2.2. Identifying Relevant Studies

A literature search was performed on four electronic databases: PubMed, Scopus, Web of Science and Ovid in March 2023 using the search string “metabolic syndrome” AND (“psoriasis vulgaris” OR “plaque psoriasis”). No additional filter was applied.

### 2.3. Study Selection

Results of the search were imported to Zotero software (Corporation for Digital Scholarship, Vienna, VA, USA)) to organise and track relevant data. This software was used to remove duplicates; document and manage the screening process; and categorise publications that meet the inclusion and exclusion criteria. Two assessors (HH and SK) independently assessed titles and abstracts based on the selection criteria. Several meetings were held over the course of screening to compare the articles against selection criteria and disagreements between assessors were resolved with a third assessor (MA), providing a binding verdict to reach a consensus. This author also performed second layer of screening to determine the final inclusion in the review. Finally, the full texts were obtained when a paper was deemed relevant. Studies were included if screening for MetS was done using published criteria, in addition to the diagnosis of psoriasis by dermatologists. Animal and child studies, reviews, case reports, proceedings, editorials, commentaries, book chapters and opinion papers were excluded. Only full-text articles were included in the final analysis.

### 2.4. Charting the Data

Data extraction was performed by HH and cross-checked by SK by tabulating it using a standardised form. Data extraction included essential characteristics such as author(s), title, journal, publication year, country of the first author, aims/objectives, sample size, participant characteristics, study design, MetS definition criteria, nutritional assessment methods and primary and secondary outcomes. Poor nutritional status was determined when an abnormal marker of nutritional status was reported. Clinical outcomes of interest were related to the presence of MetS and all key findings and outcomes were described in narrative form.

### 2.5. Collating, Summarising and Reporting the Results

Sociodemographic criteria for defining MetS, type of nutritional assessment, nutritional status and major outcomes were summarised and reported.

## 3. Results

Based on the initial search, a total of 1693 articles were identified using PubMed, Web of Science, Ovid and Scopus. A total of 271 were removed due to duplication. The remaining 1422 titles and abstracts were further screened for eligibility and 76 full texts were assessed. These were then screened according to the exclusion criteria, and finally 21 articles were included in the analysis (Figure 2).

### 3.1. Study Characteristics

A total of 21 studies were included in this review, published between 2015 and 2023. Study characteristics are presented in Table 1. A total of 61.9% of the studies were from Asian countries [28,29,30,31,32,33,34,35,36,37,38,39,40], and 52.4% were case-control studies [29,31,33,37,38,41,42,43,44,45,46] with a total of 4286 participants. Eight studies matched participants with age and sex [31,33,34,37,38,42,43,44], three studies matched for BMI [35,41,45] and only one study matched for ethnicity [45]. A total of 71.4% of studies had a higher number of males compared to females [28,29,30,31,32,33,34,35,37,38,40,41,44,46,47]. In total, 52.4% of patients with psoriasis were between 40 and 50 years old [30,31,32,35,36,38,39,41,44,46,47]. A total of 61.9% of the studies reported a mild psoriasis severity (PASI < 10) [30,31,32,33,34,35,36,38,40,41,43,45]. Nine out of twenty-one studies reported a disease duration of more than 10 years [36,39,40,41,42,43,45,47,48].

### 3.2. Metabolic Syndrome Screening

There were four MetS definition criteria used to define MetS in the included studies, as shown in Table 2. In total, 57.1% of the studies used NCEP ATP II criteria [28,29,31,32,33,35,37,38,39,42,43,48], and 33.3% were using International Diabetes Federation (IDF) criteria for MetS [30,34,36,38,40,41,44]. Two studies used the harmonised guideline [45,47], and one study used the Polish Forum of Prevention criteria [46]. The prevalence of MetS in patients with psoriasis ranged from 13.2% [36] to 67.80% [37].

### 3.3. Nutritional Assessment in Patients with Psoriasis

The waist circumference, fasting blood glucose, HDL, TG and blood pressure were the most common assessments conducted in MetS patients.

Although waist circumference was measured in all studies, two studies did not report any values [29,35]. Variations in the cut-offs for the waist circumference were observed. Eight studies used the cut-off of 102 cm for males and 88 cm for females based on the NCEP ATP III criteria [28,29,33,36,39,42,43,47]. Six studies used a cut-off for Asians, which was 90 cm for males and more than 80 cm for females [30,31,32,34,37,38]. Meanwhile, another six studies used a cut-off of 94 cm for males and 80 cm for females [40,41,44,45,46,48]. Only one study from India used an Asian Indian cut-off of 85 cm for males and 80 cm for females [35].

A total of 13 studies also reported abdominal adiposity in percentages ranging from 26.4% [33] to 78.6% [39]. Seven studies reported that more than half of the subjects had a higher waist circumference [34,39,41,42,43,47,48]; meanwhile, six studies reported that less than 50% of the subjects had a higher waist circumference [30,31,33,37,38,44].

Although all studies measured fasting blood glucose, one did not report the results [48]. Elevated blood sugar was defined as more than 100 mg/dL or 6.1 mmol/L in all the studies. Fourteen studies reported mean or median fasting blood glucose levels between 85.80 mg/dL [40] and 140.09 mg/dL [29]. Twelve studies presented the data as the prevalence of elevated blood glucose ranging from 5.3% [35] to 42.1% [33] in patients with psoriasis.

Fasting TG were assessed in all studies. Elevated fasting TG were defined as more than 150 mg/dL or 1.7 mmol/L. Sixteen studies reported mean and median fasting TG between 90.5 mg/dL [32] and 203.68 mg/dL [39]. Meanwhile, 13 studies reported a prevalence of elevated fasting TG between 26.05% [44] and 83.0% [39] in patients with psoriasis.

All studies measured and reported fasting HDL cholesterol. Low HDL was defined as less than 40 mg/dL or 1.0 mmol/L in males and less than 50 mg/dL or 1.3 mmol/L in females. Fourteen studies reported a prevalence of low HDL cholesterol in patients between 18.0% [43] and 70% [38].

Although blood pressure was measured in all studies, only three studies reported the results [29,35,40]. High blood pressure or hypertension was defined as systolic blood pressure of more than 130 mmHg or diastolic blood pressure of more than 85 mmHg. Fourteen studies reported the prevalence of hypertension in patients with psoriasis between 18.95% [37] and 61% [47]. Nine studies reported mean systolic blood pressure between 120 mmHg [32] and 141 mmHg [45] and diastolic blood pressure between 70 mmHg [32] and 90.7 mmHg [33].

### 3.4. Nutritional Status of Plaque Patients with Psoriasis (Table 3)

Nineteen studies assessed BMI [28,29,30,31,32,33,34,36,38,39,40,41,42,43,44,45,46,47,48]. Nine out of 12 Asian studies reported a mean BMI of 24.0 kg/m^2^ [33] to 31.15 kg/m^2^ [39], which falls under the overweight and obese category using the Asia–Pacific cut-off. Meanwhile, five out of seven non-Asian studies reported a mean BMI of 26.24 kg/m^2^ [42] to 31.6 kg/m^2^ [45], which is defined as the overweight and obese category as per WHO guidelines. The BMI of patients with psoriasis was higher compared to controls. Other anthropometry assessments conducted were the hip circumference in two studies [45,47] and waist-to-hip ratio in two studies [45,46].

**Table 3 nutrients-15-02707-t003:** Nutrition assessment tools/methods and nutritional status.

Author (Year), Location and Study Design	Participant Characteristics (Cases)	Participant Characteristics (Controls)	Nutritional Status	Anthropometry Measure	Biochemical Parameter	Clinical Parameters
Gisondi et al., 2007 [43] Italy Case-control	Mean age: 62.1 ± 15.1 yrs Sex: 160/178 Sample size: (n = 338) MetS: 30.1%	Mean age: 63.8 ± 20.4 yrs Sex: 150/184 Sample size: (n = 334) MetS: 20.6%	Overweight 27.7 ± 4.8 kg/m^2^	WC	x	BP
Pereira et al., 2011 [35] India Case-control	Mean age: 46.51 ± 13.53 yrs Sex: 62/15 Sample size: (n = 77) MetS: 18.2% PASI: 4.61 ± 4.18	Mean age: 43.24 ± 12.33 yrs Sex: 63/29 Sample size: (n = 92) MetS: 17.4%	x	WC, WHR	OGTT, fasting insulin, post-prandial insulin, TC, HDL, TG, FBG	BP
Damevska et al., 2013 [42] Macedonia Case-control	Mean age: 51.52 ± 15.56 yrs Sex: 52/70 Sample size: (n = 122) MetS: 24.6% PASI: 14.75 ± 12.78	Mean age: 51.56 ± 15.72 yrs Sex: 52/70 Sample size: (n = 122) MetS: 22.9%	Severe obesity PsO: 4.1% Control: 1.6% Obesity PsO: 16.4% Control: 9.8%	WC	x	BP
Akcali et al., 2014 [28] Turkey Case-control	Mean age: 49.51 ± 18.26 yrs Sex: 26/24 Sample size: (n = 50) MetS: 50%	Mean age: 47.87 ± 16.43 yrs Sex: 20/20 Sample size: Control (n = 40) MetS: 25%	Overweight PsO: 26.92 ± 4.11 kg/m^2^ Control: 25.73 ± 5.89 kg/m^2^	WC	TG, HDL, FBG, homocysteine, fibrinogen, adiponectin	BP
Baeta et al., 2014 [48] Brazil Cross sectional cohort	Mean age: 51.5 ± 14 yrs Sex: 93/97 Sample size: (n = 190) MetS: 44.9% PASI: 3.4 ± 3.03	x	Normal: 35.7% Overweight: 31.1% Obese I: 22.6% Obese II: 7.4% Obese III: 3.2%	WC	TC, LDL, HDL, TG	BP
Puig et al., 2014 [47] Spain RCT	Mean age: 43.9 ± 12.7 yrs Sex: 190/273 Sample size: (n = 273) MetS: 42% PASI: 21.2 ± 9.4	x	Overweight Male: 28.3 ± 4.6 kg/m^2^ Female: 29.6 ± 7.5 kg/m^2^	WC, HC	FBG, fasting insulin, HbA1c, TC, HDL, LDL, TG, ApoA1, ApoB, ApoA, adiponectin	BP
Ucak et al., 2014 [40] Turkey Case-control	Mean age: 32.24 ± 7.54 yrs Sex: 11/14 Sample size: (n = 25) MetS: 40% PASI: 1.96 ± 0.84	Mean age: 31.40 ± 6.77 yrs Sex: 14/11 Sample size: (n = 25) MetS: 36%	Overweight PsO: 24.96 ± 2.53 kg/m^2^ Normal Control: 23.80 ± 1.50 kg/m^2^	WC	FBG, LDL. HDL, TC, TG, HbA1c, insulin, c-peptide levels, TSH, T3, T4, ghrelin	BP
Albareda et al., 2014 [41] Spain Case-control	Mean age: 49.32 ± 13.47 yrs Sex: 55/47 Sample size: (n = 102) MetS: 52.9% PASI: 6.4 (0–36.6)	Mean age: 48.71 ± 13.84 yrs Sex: 55/47 Sample size: (n = 102) MetS: 34.3%	Overweight PsO: 27.7 (18.9–41.79) kg/m^2^ Control: 27.36 (18.24–40.5) kg/m^2^	BMI, WC	x	BP
Irimie et al., 2015 [44] Romania Case-control	Mean age: 49.51 ± 18.26 yrs Sex: 75/67 Sample size: (n = 142) MetS: 13.4%	Mean age: 47.87 ± 16.43 yrs Sex: 88/79 Sample size: (n = 167) MetS: 10.8%	x	WC	TG, TC, HDL, LDL	BP
Kothiwala et al., 2016 [33] India Case-control	Mean age: 37.9 ± 13.26 yrs Sex: 102/38 Sample size: (n = 140) MetS: 39.3%	Mean age: 36.1 ± 11.63 Sex: 97/43 Sample size: (n = 140) MetS: 17.1%	Normal PsO: 24.0 ± 4.43 kg/m^2^ Control: 22.6 ± 3.71 kg/m^2^	BMI, WC	x	Carotid intima media thickness (CIMT) and BP
Pongpit et al., 2016 [36] Thailand Cross-sectional cohort	Mean age: 49.2 ± 14.0 yrs Sex: 75/90 Sample size: (n = 165) MetS: 50.3% PASI: 3.0 ± 2.7	x	Overweight 24.8 ± 4.7 kg/m^2^	BMI, WC	FBG, total cholesterol, TGL, LDL, HDL, AST, ALT	BP
Sharma et al., 2016 [38] India Case-control	Mean age: 44.94 ± 11.08 yrs Sample size: (n = 100) MetS: 38%	Mean age: 43.28 ± 12.06 yrs Sample size: (n = 100) MetS: 12%	Overweight: 50% Obese: 50%	WC	x	x
Uczniak et al., 2016 [46] Poland Case-control	Mean age: 46 ± 13 yrs Sex: 138/108 Sample size: (n = 246)	Mean age: 46 ± 13 yrs Sex: 35/40 Sample size: (n = 75)	x	WC, WHR	x	BP
Bulur et al., 2017 [29] Turkey Case-control	Mets Mean age: 39.00 ± 13.59 yrs Sex: 23/16 Sample size: (n = 39) No MetS Mean age: 46.95 ± 10.68 9 yrs Sex: 11/10 Sample size: (n = 21) MetS: 65%	Mean age: 36.73 ± 10.07 yrs Sex: 7/8 Sample size: (n = 12)	Overweight PsO with MetS: 26.35 ± 4.92 kg/m^2^ Control: 25.56 ± 2.92 kg/m^2^ Obese PsO without MetS: 33.55 ± 5.16 kg/m^2^	WC	FBG, TG, TC, LDL, HDL	BP
Girisha et al., 2017 [31] India Case-control	Mean age: 45.5 ± 12.6 yrs Sex: 118/38 Sample size: (n = 156) MetS: 28.8%	Mean age: 45.4 ± 12.5 yrs Sex: 118/38 Sample size: (n = 156) MetS: 16.7%	Normal PsO 24 ± 4.8 kg/m^2^ Overweight Control 25.2 ± 2.2 kg/m^2^	BMI, WC	x	BP
Salunke et al., 2017 [37] India Case-control	Mean age: 36.88 ± 13.37 yrs Sex: 71/24 Sample size: (n = 95) MetS: 38.9%	Mean age: 36.30 ± 13.07 yrs Sex: 73/22 Sample size: (n = 95) MetS: 21.05%	x	WC	FBS, TG, HDL	BP
Korkmaz et al., 2018 [32] Turkey Case-control	MetS Mean age: 44.2 ± 10.1 yrs Sex: 18/20 Sample size: (n = 38) PASI: 2.51 ± 1.90 No MetS Mean age: 43 ± 9.3 yrs Sex: 22/16 Sample size: (n = 38) PASI: 30.04 ± 2.18 MetS without PsO Mean age: 45.8 ± 9.7 yrs Sex: 19/19 Sample size: (n = 38)	Mean age: 42 ± 10.4 yrs Sex: 18/17 Sample size: (n = 35)	Obese MetS: 34 ± 9.3 kg/m^2^ Overweight PsO with MetS: 29.9 ± 6.6 kg/m^2^ PsO: 28 ± 4.2 kg/m^2^ Normal Control: 25 ± 5.4 kg/m^2^	BMI, WC	FBG, urea, creatinine, TC, HDL, LDL, TG, TSH, ft3, ft4, platelets	BP
Mahyoodeen et al., 2019 [45] South Africa Case-control	Mean age: 53.3 ± 14.5 yrs Sex: 48/55 Sample size: (n = 103) MetS: 52.4% PASI: 4.80 (2.40–11.7)	Mean age: 47.9 ± 14.5 yrs Sex: 37/61 Sample size: (n = 98) MetS: 33.7%	Obesity PsO: 31.6 ± 8.42 kg/m^2^ Vitamin D deficiency 19.0 (12.6, 24.1) Overweight Control: 29.3 ± 6.67 kg/m^2^ Vitamin D deficiency 18.1 (13.1, 24.3)	BMI, WC, HC, WHR	FBG, HbA1c, TC, TG, HDL, LDL, TG, lipoprotein A, urea, creatinine, CRP, serum vitamin D	BP
Tas et al., 2021 [39] Turkey Cross sectional cohort	Mean age: 46.5 ± 13.2 yrs Sex: 53/59 Sample size: (n = 112) MetS: 67.8%	x	Obesity PsO with MetS: 31.15 ± 4.88 kg/m^2^ Overweight PsO without MetS: 28.33 ± 4.08 kg/m^2^	WC	x	x
Deoghare et al., 2022 [30] India Cross-sectional cohort	Mean age: 40.45 ± 12.42 yrs Sex: 23/9 Sample size: (n = 32) MetS 37.5% PASI: 8.63 ± 7.49	x	x	WC	FBS, TG, HDL	BP, carotid intima thickness, epicardial fat thickness
Patil et al., 2022 [34] India Case-control	Mean age: 39.67 ± 12.36 yrs Sex: 26/16 Sample size: (n = 42) MetS: 36% PASI: 6.42 ± 4.16	Mean age: 39.71 ± 12.32 yrs Sex: 26/16 Sample size: (n = 42) MetS: 24%	Overweight and vitamin D deficiency: 26.96 ± 5.67 kg/m^2^ Overweight and insufficient vitamin D: 26.06 ± 5.98 kg/m^2^	WC	FBG, TG, HDL, serum 25(OH)D levels	BP

Abbreviations: ALT: alanine transaminase, AST: aspartate aminotransferase, ApoA1: apolipoprotein A1, ApoA: apolipoprotein A, ApoB: apolipoprotein B, BP: blood pressure, BMI: body mass index, CRP: c-reactive protein, FBG: fasting blood sugar, ft3: free triiodothyronine, ft4: free thyroxine, HDL: high-density lipoprotein, LDL: low-density lipoprotein, MetS: metabolic syndrome, OGTT: oral glucose tolerance, PASI: psoriasis area severity index, PsO: patients with psoriasis, TSH: thyroid-stimulating hormones, T3: triiodothyronine, T4: thyroxine, TC: total cholesterol, TGL: triglycerides, WC: waist circumference, WHR: waist–hip ratio, yrs: years, 25 (OH) D: 25-hydroxy vitamin D.

### 3.5. Other Additional Biomarkers Assessed

Two studies that measured patient HbA1c levels found these to be elevated [45,47]. Meanwhile, an oral glucose tolerance test (OGTT) was performed in two studies [35,41]. Fasting insulin was also measured in six studies and used to calculate insulin resistance using the homeostasis model assessment of insulin resistance (HOMA-IR) formula [35,36,39,40,41,47]. Only one study measured post-prandial insulin. A total of 11 studies measured total cholesterol [29,31,32,35,36,40,41,44,45,47,48], and 10 studies measured low-density lipoprotein (LDL) [29,32,35,36,40,41,44,45,47,48]. Three studies reported a prevalence of elevated total cholesterol between 18.5% [48] and 37.7% [35] in patients with psoriasis. Mean and median total cholesterol levels in seven studies ranged from 176.38 [29] to 223.42 mg/dL [44]. Three studies reported a prevalence of elevated LDL cholesterol between 16.9% [48] and 50.6% [35] among patients with psoriasis. Other biochemical assessments conducted were serum vitamin D levels [34,45], adiponectin [28,47], hs-CRP [45,47], apoprotein A [47], apoprotein B [47], leptin [47], homocysteine [28], fibrinogen [28], AST [36], ALT [36], platelets [32], c-peptide [40], ghrelin [40] and eGFR [45].

Other clinical assessments were the carotid intima thickness in two studies [30,33], epicardial fat thickness in one study [30] and ultrasonography of the liver in one study [36].

### 3.6. Psoriasis Area and Severity Index

Only 11 studies assessed the PASI with a score ranging from 1.9 to 21.2 [30,32,34,35,36,40,41,42,45,47,48]. Of these, three studies reported moderate scores [30,34,41], one reported a severe score [42] and two reported a very severe score [32,47].

## 4. Discussion

In this scoping review, we have summarised nutritional assessments performed in patients with psoriasis who were at risk of MetS. The nutritional assessments conducted were simple anthropometric measurements such as the weight and height to calculate BMI, waist circumference to predict abdominal adiposity, metabolic markers such as fasting blood glucose, TG, LDL and HDL cholesterol and blood pressure. These assessments were part of the MetS screening criteria to ascertain insulin resistance among patients with MetS. However, several studies also performed additional assessments to assess insulin resistance, liver impairment, kidney dysfunction and inflammation. Although most of the studies performed nutritional assessment, we were not able to deduce the nutritional status as the assessments were incomplete. Only 17 studies reported BMI results and we were not able to determine if all Asian studies used the Asia–Pacific cut-off instead of the WHO cut-off for BMI. Most patients with psoriasis suffered from poor nutritional status compared to controls. Only two studies assessed the vitamin D status among patients, which indicated vitamin D deficiency. However, no studies assessed the dietary intake of patients even though there is a possibility of obese patients being at risk of malnutrition.

Patients with psoriasis in the studies reviewed were mostly overweight or obese. An increased body mass could potentially decrease the drug distribution in the body, reducing the effectiveness of systemic or biological treatment [49,50]. In addition, a higher BMI also increases psoriasis severity [50,51,52] and induces inflammation [24,53], thus increasing the risk of comorbidities in patients.

Therefore, early detection of comorbidities and weight loss has been recommended for patients with psoriasis [54,55,56]. In the included studies, obesity was assessed by BMI, whereas abdominal obesity was assessed using the waist circumference and waist-to-hip ratio. BMI is widely used in clinical settings as it is a convenient index to determine nutritional status. However, it has its limitations, as a nutritional status determined through BMI does not distinguish the difference between fat mass and muscle mass, as it measures excess weight rather than excess fat. As BMI cannot determine the distribution of body fat [57], waist circumference is a better predictor of adiposity compared to BMI and is an important measure of central obesity, a predisposing factor for psoriasis [58]. A high volume of visceral adipose tissue (VAT) is associated with chronic inflammation [59]. White adipose tissue that is mostly formed as VAT induces inflammation and cell dysfunction, reduces insulin sensitivity and disrupts glucose and lipids by releasing proinflammatory adipokines [53,60]. However, it is recommended that both BMI and waist circumference are interpreted together to determine those who are abdominally obese with increased risks [61].

It was also observed that most studies did not classify waist circumference according to sex, leading to a possibility of misclassification. Waist circumference is also prone to measurement errors if it is not measured by trained staff. Therefore, there is potentially a need for a more reliable assessment of body composition to provide better data on the nutritional status, such as the use of bioelectrical impedance analysis (BIA), a safe, inexpensive, non-invasive technique that can distinguish the fat and muscle mass. The associated bioelectrical device employs a small electric current to estimate the resistance and reactance of various tissues in the body. Areas with high fat stores will show a high impedance reading compared to areas with high muscle stores that largely store body water [62]. It is also pertinent to assess the appendicular muscle mass due to an increased risk of sarcopenic obesity, particularly in older adults [63,64]. A study on patients with chronic plaque psoriasis found an association with myosteatosis and not sarcopenia, but the sample size in this study was small and the mean age of respondents was 45 years, which warrants further investigation in this population [65]. Meanwhile, several other studies have shown that sarcopenic obesity may also increase the risk of MetS [66,67,68], which poses a concern, stressing the importance of body composition measures.

Biochemical parameters that were mainly assessed were fasting blood glucose, HDL cholesterol and TG. Several studies also assessed the complete lipid profile including total cholesterol and LDL cholesterol. However, not all studies classified HDL cholesterol according to sex. There was a 16.6% increased risk of psoriasis in those with low HDL and the risk was higher in females, which was 16.9% [69]. Therefore, classification according to sex is necessary.

Although insulin resistance was not compulsory to define MetS, a few studies assessed this using various indexes. One of them was HOMA-IR, whereby the measured fasting insulin was multiplied by fasting blood glucose and a value of <2.5 was considered part of the normal range [70]. Although this method is simple, the values in subjects treated with insulin might not be accurate [71]. C-peptide was also used in a study to calculate the HOMA-IR and this is used interchangeably with fasting insulin as it has a longer half-life compared to insulin. A value less than 0.2 nmol/L indicates the diagnosis of type 1 diabetes mellitus and it is also a marker for microvascular and macrovascular complications [72]. However, both measures were mainly used to determine insulin resistance in patients with psoriasis [73,74].

In addition, the quantitative insulin-sensitivity check (QUICKI) was also determined by log transforming values of fasting glucose and the fasting insulin level with a cut-off of <0.33 for insulin resistance [75]. This index is consistent and precise but there are significant variations in the normal range based on the assays performed in the laboratory [71]. The OGTT was also performed in some studies to assess insulin sensitivity using the Matsuda index. The index is calculated from plasma glucose and insulin concentrations in the fasting state and during the OGTT [73]. A value of < 4.3 is used to predict insulin resistance. Although this index represents both hepatic and peripheral tissue sensitivity, it has a weak correlation with insulin resistance in those with diabetes [71]. Meanwhile, HbA1c was also assessed; values between 5.7% and 6.4% were considered markers of an increased risk for diabetes [74]. Apart from fasting insulin, HbA1c and OGTT are routinely performed to screen for diabetes and are performed in a clinical setting.

Other biochemical parameters such as apolipoprotein levels were measured to determine cardiovascular risk in patients. The ratio of apolipoprotein B (ApoB) to apolipoprotein AI (ApoAI) is a marker for lipid disturbances as ApoAI is associated with HDL particles and it can predict the risk of myocardial infarction [76,77]. As patients with psoriasis have a greater risk of developing cardiovascular diseases, ApoB has been demonstrated as a better predictor for cardiovascular disease risk compared to LDL [78,79]. Other parameters related to cardiovascular risk such as serum homocysteine, platelets and fibrinogen were also assessed. Elevated homocysteine levels have also been associated with severity of psoriasis. Meanwhile, platelet activation has been associated with endothelial dysfunction, inflammation and immune function in patients with psoriasis [80]. An increased level of fibrinogen has also been associated with severity in patients with psoriasis [81]. Meanwhile, the thyroid status was also assessed by measuring the thyroid-stimulating hormone (TSH), triiodothyronine (T4) and thyroxine (T4). T3 is the active hormone converted from T4. A high amount of produced T3/T4 will suppress TSH through negative feedback inhibition and vice versa [82]. Smoking, BMI and iodine intake have been associated with changes in TSH, whereby smoking led to hypothyroidism, high iodine intake led to hyperthyroidism and there was a positive correlation between BMI levels and TSH [83]. Meanwhile, thyroid dysfunction, particularly hypothyroidism, has been observed in patients with psoriasis [84,85].

Adiponectin and c-reactive protein (CRP) were assessed as inflammatory markers—increased levels were usually associated with the severity of psoriasis [86,87]. Meanwhile, higher levels of ghrelin were associated with an improved psoriasis severity score in a mouse model [88], and have also been assessed to determine the efficacy of biologics, which were also measured in the studies included [89]. Other biochemical markers assessing kidney functions (e.g., urea and creatinine) and liver function (e.g., alanine transaminase and aspartate aminotransferase) were also measured in several studies. Patients with psoriasis have an increased risk of developing chronic kidney failure as a result of psoriatic inflammation [90,91]. Meanwhile, non-alcoholic fatty liver disease was associated with psoriasis, particularly among men [92], as a result of insulin resistance due to an increased level of proinflammatory adipokines [93]. 25-hydroxyvitamin D was the only biochemical measure that assessed the nutritional status, as this test was able to determine the vitamin D status in the body. It has been demonstrated that patients with psoriasis are at risk of vitamin D deficiency [94,95]. As vitamin D is involved in bone growth, low levels may lead to bone loss, thereby increasing the risk of osteoporosis in these patients [96].

Clinical parameters that were assessed were mainly blood pressure, carotid intima thickness and epicardial fat thickness. Blood pressure is one of the most important clinical markers to define MetS, as hypertension is a cardiovascular risk factor and psoriasis is reported to be associated with an increased risk of hypertension [97,98]. Meanwhile, the carotid intima and epicardial fat thickness are markers for atherosclerosis and coronary artery diseases, respectively, and they are associated with visceral adiposity in patients with psoriasis [99,100].

It was observed that most of the patients with psoriasis in the studies included were either overweight or obese with abdominal adiposity. Studies that measured serum vitamin D showed that patients with psoriasis were at risk of vitamin D deficiency related to poor dietary intake or a lack of exposure to sunlight [101]. Although the aim of these studies was to determine the MetS status in these patients, some of the nutritional assessments performed were useful in determining the nutritional status of patients with psoriasis, such as the BMI, waist circumference and vitamin D status. These anthropometry methods were non-invasive, easy and quick to perform and could be conducted routinely in a clinical setting.

Besides vitamin D deficiency, patients with psoriasis could also be at risk of other nutritional deficiencies related to their dietary practices. An Italian study reported the presence of eating disorders among patients with psoriasis [102]. Meanwhile, other studies have reported an imbalance of nutrient intake related to a diet high in fat and simple carbohydrates and low in fibre [23,24]. Dietary intake is related to psoriasis severity whereby a diet high in saturated fat and red and simple sugar is associated with inflammation. Meanwhile, vitamin D, vitamin B12, vitamin A and selenium play an important role in improving severity [6,103,104]. Therefore, dietary assessment is pertinent to determine the nutritional status of patients with psoriasis. Several dietary assessment methods can be used to determine the dietary intake of patients with psoriasis. Trained dietitians or researchers could conduct a quick 24-h multiple-pass recall by asking open-ended questions to patients to enquire about actual intake in a specific period. Although this recall may rely on the patient’s ability to recall intake, a trained individual may be able to prompt patients to gather sufficient information in multiple passes such as dietary recall for the past 24-h, followed by details of food and beverages consumed, the portion size or ingredients and a summary of the entire day’s intake. This method is used daily in busy clinical practices by dietitians or nutritionists [105,106]. Meanwhile, in the research setting, food frequency questionnaires or food records could be employed. A food frequency questionnaire is used to estimate the usual intake over 6 months to a year. It is a closed method using a questionnaire and may be subjected to recall bias. However, it is still considered a reliable tool for dietary intake and is used in epidemiological studies [106,107]. A food record or diary is a very useful tool in research whereby patients could record their daily intake for a period of 3, 4 or 7 days with the help of a guide to assist them in recording their food and beverages. Although it is time-consuming and there is a tendency to under-report or over-report, this tool could capture current intake, timing or meals, frequency or eating out. Cooking methods are also very useful in assessing nutrient deficiencies and trained researchers could clarify missing information through a debriefing process [106,108]. Although each of these proposed methods has its limitations, these can be minimised either by skilled professionals or by the use of technology to assist reporting.

Patients with psoriasis who are overweight and obese are more likely to develop MetS. Therefore, early dietary intervention is needed to improve the nutritional status, reduce the risk of metabolic disorders and improve the effectiveness of treatments [103,104,109,110]. Dietary recommendations are often individualised based on the patients’ comorbidities. Obese or overweight patients with psoriasis without MetS would usually be advised on caloric restriction and weight reduction. A low-calorie diet lowers BMI and reduces inflammation and oxidative stress, thus improving the response to treatment [54,110]. Meanwhile, patients with MetS will require a more in-depth dietary consultation based on their elevated biochemical and clinical parameters. Besides caloric restriction and weight reduction, recommendations also include the restriction of total fat and refined carbohydrates. A Mediterranean diet is more appropriate for these patients as it is a balanced diet, as well as high in antioxidants, polyphenols and mono-unsaturated fats. The anti-inflammatory effect of the diet has been associated with reducing coronary heart disease, increasing immunity and reducing the inflammatory response [103,111]. Dietary supplementation, particularly with vitamin D, could also be recommended for patients with deficiency [103].

Although the severity of psoriasis has been associated with MetS in several studies, it is not routinely assessed in clinical practice. The PASI is commonly used to determine the efficacy of treatment in clinical trials, as scores reflect improvement or worsening of the condition. The scoring is based on the redness, thickness and scaliness of target plaques on the head, upper limbs, trunk and lower limbs. The scoring is from 0 to 6 and is based on the percentage of skin or body surface area (BSA) covered with psoriasis. A score of 0 indicates no involvement, 1 = <10% BSA, 2 = 10–29% BSA, 3 = 30–49% BSA, 4 = 50–69% BSA, 5 = 70–89% BSA and 6 = 90–100% BSA [4]. However, the BSA is further categorised as mild (0–5%), moderate (>5–10%), severe (>10–15%) and very severe (>15%) [112]. Psoriasis severity has been associated with MetS whereby overexpression of proinflammatory cytokines leads to inflammation, causing prolonged tissue damage, increasing the risk of comorbidities such as obesity, MetS, non-alcoholic fatty liver disease, psoriatic arthritis and inflammatory bowel disease [113,114,115]. However, no significant association between severity and MetS was reported in any of the studies reviewed, which is inconsistent with published meta-analyses [113,116].

It is also important to note that psoriasis begins in childhood in almost one-third of cases. With obesity, insulin resistance and MetS developing in even younger children of pre-pubertal ages, early screening and detection are necessary [117,118]. Obesity, particularly central obesity, is more prevalent in children with psoriasis and is usually assessed using the waist circumference percentile, weight-to-height ratio and body mass percentile [117,118,119]. Several other studies have also observed insulin resistance [117,119] and cardiovascular comorbidities such as hypertension in children [120,121,122]. However, studies assessing MetS in children are still scarce compared to adults, possibly due to a lack of standardised international guidelines to screen children for MetS. As treatment options are limited for children, dietary intervention should be commenced as early as possible.

This scoping review summarised the nutrition assessment and nutritional status of patients with psoriasis with MetS. Although we were able to address the research gaps, there are also several limitations in this review. Firstly, only studies that assessed MetS using published criteria were included, thereby limiting the number of studies in the review. Secondly, studies with self-reported psoriasis were not included in this review. Therefore, this may have excluded some relevant studies that may have assessed the nutritional status of patients with psoriasis.

## 5. Conclusions

In conclusion, patients with psoriasis are at risk of MetS and an increase in body mass and adiposity may play a role. Although anthropometry measurement is employed to determine the nutritional status, it is also evident that patients with psoriasis may have a risk of nutrient deficiencies, which is not assessed in clinical practice or research settings. Additional assessments, such as body composition and dietary intake, need to be performed to determine the nutritional status of these patients and assess their health risk, and subsequently devise a suitable intervention. This review provides practical insights for an integrative collaboration between dermatologists and dietitians for better treatment and intervention.

## Figures and Tables

**Figure 1 nutrients-15-02707-f001:**
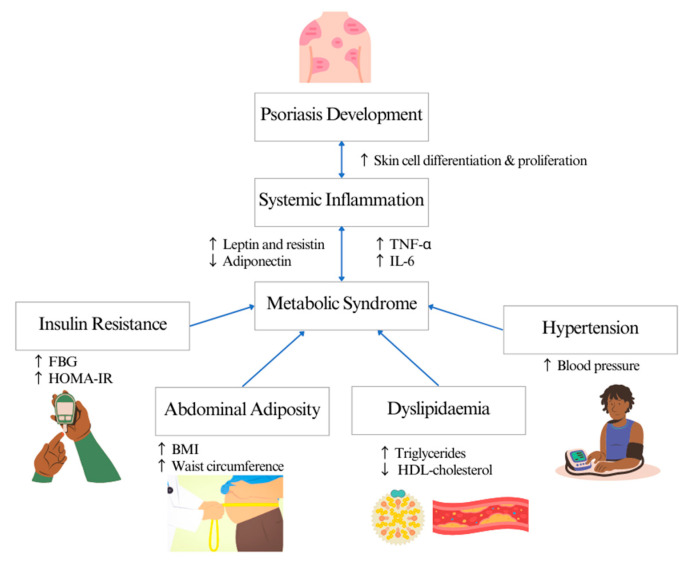
Systematic inflammation linking psoriasis and metabolic syndrome.

**Figure 2 nutrients-15-02707-f002:**
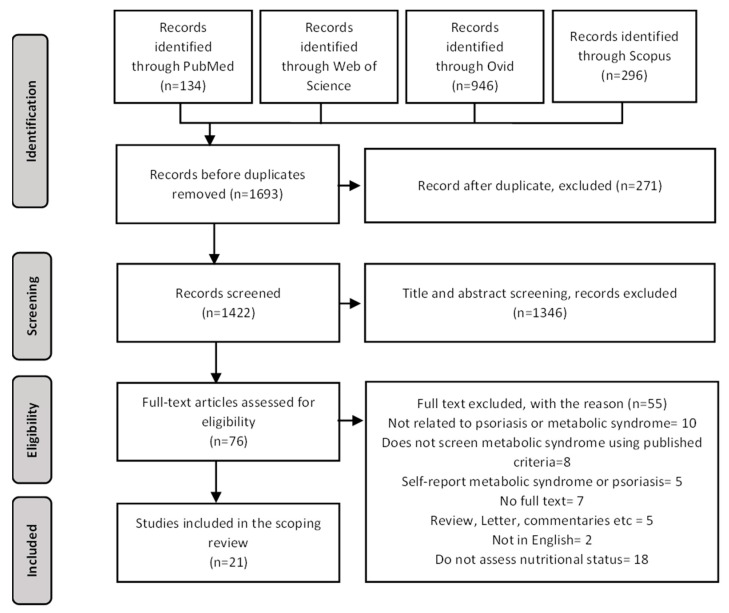
Flow chart of search strategies.

**Table 1 nutrients-15-02707-t001:** Study characteristics (*n* = *21*).

Characteristics		*n* (%)
Year	2005–2009	*1* (4.76)
	2010–2014	*7* (33.3)
	2015–2019	*10* (47.6)
	2020–2023	*3* (14.3)
Continent	Africa	*1* (4.8)
	Asia	*13* (61.9)
	Europe	*6* (28.6)
	South America	*1* (4.8)
Study design	Case-control	*11* (52.4)
	Cross-sectional	*8* (38.1)
	Cross-sectional, case-control	*1* (4.8)
	RCT	*1* (4.8)

**Table 2 nutrients-15-02707-t002:** Metabolic syndrome screening criteria.

Metabolic Syndrome Definition	*n* (%)
NCEP ATP III	*12* (57.1)
IDF	*7* (33.3)
Harmonised	*2* (9.5)
Polish Forum of Prevention	*1* (4.8)

## Data Availability

Data sharing is not applicable to this article as no datasets were generated or analysed during the current study.

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
