# Peer review of "Metabolic Syndrome Screening and Nutritional Status of Patients with Psoriasis: A Scoping Review"

_nutrients, 2023, doi:10.3390/nu15122707_

Round 1
Reviewer 1 Report
Dear Authors,
The manuscript titled “Metabolic syndrome screening and nutritional status of psoriasis patients: A scoping review” by Nur Hanisah Mohamed Haris Shanthi Krishnasamy, Chin Kok Yong, Vanitha Mariappan and Mohan Arumugam – Nutrients 2414737 identifies and summaries metabolic syndrome screening criteria and the tools/ methods used in nutrition assessment in plaque psoriasis patients.
1. The paper lacks graphs showing the relationship between analyzed metabolic parameters and psoriasis development.
2. Is there a relationship between skin changes (plaque psoriasis surface area) and metabolic disorders?
3. The authors show the relationship between psoriasis and BMI. What kind of nutritional recommendations can you give patients with psoriasis? Are they identical, like in metabolic syndrome or specific to psoriasis?
-
Author Response
Point 1:The paper lacks graphs showing the relationship between analyzed metabolic parameters and psoriasis development.
Response 1:
Thank you for your comment and the valuable feedback. We have explained the relationship between the reported parameters and psoriasis and inserted a figure. Please refer to line 57-65 and Figure 1.
Point 2: Is there a relationship between skin changes (plaque psoriasis surface area) and metabolic disorders?
Response 2:
Thank you for your question. Nutritional recommendations for psoriasis patients is usually individualised and they may differ depending on their nutritional status & comorbidities. Patients who are obese without metabolic syndrome are usually advised on caloric restriction and physical activity for weight loss. Meanwhile patients with metabolic syndrome would need to restrict total fat and refined sugar besides caloric restriction and lifestyle changes. As these patients may also be at risk of Vit D deficiency, supplementation will also be recommended. We have included dietary recommendations in our manuscript as well. Please refer to line number 414-429.
|
Reviewer 2 Report
This is a very interesting review regarding the prevalence of metabolic syndrome in patients with psoriasis. Patients with psoriasis have high risk of cardiovascular and metabolic comorbidities, such as hyperlipidemia, diabetes mellitus, hypertension, obesity, central obesity and metabolic syndrome.
In this study, the nutritional status of patients with psoriasis is also reviewed and discussed.
The review about metabolic syndrome and nutritional status is methodologically well-conducted and the results are also well described.
In the clinical practice, the early detection of cardiovascular and metabolic comorbidities is crucial in patients with inflammatory skin diseases, such as psoriasis. In fact, the early detection of these diseases, could be allow patients to avoid the use of pharmacological treatments, acting in other aspects of life (changing life-styles, increasing sport activities and improving mood of alimentation).
Regarding this topic, the authors should be also highlighted in the section of Discussion the importance to early detect cardiovascular and metabolic comorbidities, especially in children and adolescents. As a matter of fact, central obesity and over-weight and obesity in pediatric age, are recognized factors which can increased the risk of develop other cardiovascular comorbidities in adult age (such as metabolic syndrome, hypertension, hyperlipidemia, but also myocardial infarction or stroke), probably through systemic inflammation and insulin resistance, causing endothelial dysfunction.
The authors should highlight these aspects in the Discussion, also referring to the several studies that have reported the increase of cardiovascular and metabolic diseases, not only in adults, but also in children with psoriasis, such as “Metabolic syndrome and insulin resistance in pre-pubertal
children with psoriasis” by Caroppo F. et al. (2021), “Central obesity in children with psoriasis” by Guidolin L. et al. (2018), “Psoriasis and metabolic and cardiovascular comorbidities in children: A systematic review” by Badaoui A. et al. (2018), “Psoriasis and metabolic syndrome in children: current data” by Pietrzak A. et al. (2017). In clinical practice, these aspects are important, for a complete therapeutic education for children with psoriasis and their parents.
Reference section is adequate and up-to-date. Some mistakes in the English language are present, the text should be revised for a further check of English language.
Some mistakes in the English language are present, the text should be revised for a further check of English language.
Author Response
Point1:
This is a very interesting review regarding the prevalence of metabolic syndrome in patients with psoriasis. Response 1: Thank you very much for your feedback. Point 2: Patients with psoriasis have high risk of cardiovascular and metabolic comorbidities, such as hyperlipidemia, diabetes mellitus, hypertension, obesity, central obesity and metabolic syndrome. In this study, the nutritional status of patients with psoriasis is also reviewed and discussed. The review about metabolic syndrome and nutritional status is methodologically well-conducted and the results are also well described. Regarding this topic, the authors should be also highlighted in the section of Discussion the importance to early detect cardiovascular and metabolic comorbidities, especially in children and adolescents. As a matter of fact, central obesity and over-weight and obesity in pediatric age, are recognized factors which can increased the risk of develop other cardiovascular comorbidities in adult age (such as metabolic syndrome, hypertension, hyperlipidemia, but also myocardial infarction or stroke), probably through systemic inflammation and insulin resistance, causing endothelial dysfunction. The authors should highlight these aspects in the Discussion, also referring to the several studies that have reported the increase of cardiovascular and metabolic diseases, not only in adults, but also in
children with psoriasis, such as “Metabolic syndrome and insulin resistance in pre-pubertal children with psoriasis” by Caroppo F. et al. (2021), “Central obesity in children with psoriasis” by Guidolin L. et al. (2018), “Psoriasis and metabolic and cardiovascular comorbidities in children: A systematic review” by Badaoui A. et al. (2018), “Psoriasis and metabolic syndrome in children: current data” by Pietrzak A. et al. (2017). In clinical practice, these aspects are important, for a complete therapeutic education for children with psoriasis and their parents. The authors should highlight these aspects in the Discussion, also referring to the several studies that have reported the increase of cardiovascular and metabolic diseases, not only in adults, but also in children with psoriasis, such as “Metabolic syndrome and insulin resistance in pre-pubertalchildren with psoriasis” by Caroppo F. et al. (2021), “Central obesity in children with psoriasis” by Guidolin L. et al. (2018), “Psoriasis and metabolic and cardiovascular comorbidities in children: A systematic review” by Badaoui A. et al. (2018), “Psoriasis and metabolic syndrome in children: current data” by Pietrzak A. et al. (2017). In clinical practice, these aspects are important, for a complete therapeutic education for children with psoriasis and their parents. Response 2: Thank you very much for providing us with the references. We have addressed development of metabolic syndrome in children with psoriasis based on the feedback provided. We agree that children with psoriasis also have an increased risk to develop metabolic syndrome. We have added a section on children based on the references provided in our manuscript. Please refer to line number 448-458. Point 3: Reference section is adequate and up-to-date. Response 3: Thank you for your comments. Point 4: Comments on the Quality of English Language Some mistakes in the English language are present, the text should be revised for a further check of English language. Response 4: Thank you for your valuable feedback, we have edited our manuscript for proper English, grammar, punctuation and spelling.
|